# Drainage and refill of an Antarctic Peninsula subglacial lake reveals an active subglacial hydrological network

Dominic A. Hodgson[1], Tom A. Jordan[1], Neil Ross[2], Teal R. Riley[1], Peter T. Fretwell[1]

[1] British Antarctic Survey, High Cross, Madingley Road, Cambridge CB3 0ET, UK

[2] Newcastle University, Claremont Road, Newcastle Upon Tyne, NE1 7RU, UK

*Correspondence to*: Dominic A Hodgson (daho@bas.ac.uk)

**Abstract.** The presence of subglacial lakes and subglacial hydrological networks under the East and West Antarctic Ice Sheets is now relatively well understood, whilst their influence on ice dynamics is the subject of ongoing research. In contrast, little is known about subglacial lakes and hydrological networks under the

Antarctic Peninsula Ice Sheet and how these are influencing glacier behaviour. Here we describe the rapid drainage and slow refill of a subglacial lake under Mars Glacier using remote sensing and aerogeophysics. Results suggest drainage of the subglacial lake occurred prior to 2011, resulting in collapse of the overlying ice into the newly formed subglacial cavity. The cavity has been refilling since this time, with peak rates of infilling associated with seasonal meltwater activity. We review evidence for similar features elsewhere in the Antarctic

Peninsula and discuss whether their appearance marks a threshold shift in the thermal regime of the glaciers and the activation or enhancement of their subglacial hydrological networks by surface meltwater. Collectively, these features show coupling of climate processes and the bed of the region's glaciers highlighting their ongoing vulnerability to climate change.

## 1 Introduction

Changes in glacial and subglacial hydrology can result in significant impacts on glacier and ice sheet dynamics (Ashmore and Bingham, 2014), including decreases in basal friction and short-term accelerations in ice flow (Bartholomew et al., 2012). Active subglacial hydrological networks have been revealed from changes in ice surface elevation detected by satellite altimetry and GPS (Willis et al., 2015; Joughin et al., 2016; Siegfried and Fricker, 2018; Neckel et al., 2021; Livingstone et al., 2022). Radio Echo Sounding has been used to detect the

presence of subglacial lakes (Siegert et al., 2005) and over snow seismic surveys have been used to characterise their water column, and sediment properties (Rivera et al., 2015; Smith et al., 2018). Much of this work has been carried out on large subglacial lakes under the East and West Antarctic Ice Sheets, while small lakes and hydrological networks under the valley glaciers of the Antarctic Peninsula Ice Sheet have received comparatively little attention (cf. the study of Alaskan 'alpine subglacial lakes' by Capps et al., 2010). In a recent

global inventory of subglacial lakes, only one is included on the Antarctic Peninsula (Livingstone et al., 2022) beneath Crane Glacier in northeast Graham Land (Scambos et al., 2011, Fig. 1). Another has previously been described and directly sampled on southern Alexander Island (Hodgson et al., 2009a; Hodgson et al., 2009b; Pearce et al., 2013).

Rapid warming of the Antarctic Peninsula region over the last five decades had led to its ice caps and valley

glaciers losing mass at an average rate of 24 Gt yr$^{-1}$, contributing $2.5 \pm 0.4$ mm sea-level rise since 1979

(Rignot et al., 2019). Associated with this mass loss there has been widespread accumulation of seasonal surface meltwater, which models predict will double in volume by 2050 (Trusel et al., 2015). This meltwater is particularly widespread at lower altitudes, for example on ice shelves (Kingslake et al., 2017), in glacier ablation zones, and at glacier margins and shear zones (e.g., Figs. 1b and d) and in areas where proximity to overland meltwater, increased radiative melt from low albedo rock outcrops, and the supply of surface dust act to increase meltwater volume. In some years, sustained positive air temperatures have resulted in exceptional ice shelf meltwater accumulations (Banwell et al., 2021). This water refreezes in winter, percolates into fractures, or drains into the underlying ocean through moulins or ice dolines (cf. Bindschadler et al., 2002; Lenaerts et al., 2017; Warner et al., 2021). These processes have contributed to ice shelf collapse due to meltwater driven fracture (Van Den Broeke, 2005; Lai et al., 2020).

Whilst the presence of seasonal meltwater on Antarctic Peninsula ice shelves is well-documented and monitored, the role of water on the surface, within and at the base of the region's glaciers is less well studied (Tuckett et al., 2019), which limits our understanding of regional glacier dynamics. In this paper we use remote sensing and aerogeophysical data to report and describe the rapid drainage of a previously unknown subglacial lake under Mars Glacier on Alexander Island which caused collapse of the overlying ice into a subglacial cavity. We review evidence for similar features elsewhere in the Antarctic Peninsula and Antarctica and discuss whether their appearance marks a shift in the thermal regime of (some) Antarctic glaciers and the activation or enhancement of their subglacial hydrological networks.

**1.1 Site description**

The main study site is an ice depression located on the western side of Mars Glacier, south-eastern Alexander Island (71°51'01''S, 68°28'36''W) (Figs. 1-3). The depression is situated within a broader cirque occupied by ice, snow, and bedrock outcrops at the northern end of Phobos Ridge. It appears to be the result of a vertical collapse of the ice surface, and we refer to it here as the Phobos Ice Collapse Structure (PICS). Mars Glacier is 15 km long and 3-4 km wide and lies between Two Step Cliffs and Phobos Ridge. It feeds into the lower reaches of Saturn Glacier which discharges into George VI Ice Shelf across an ice shelf shear zone (Fig. 1b). Phobos Ridge consists of sandstones and shales and has three small west-east oriented ice filled valleys or cirques which are part of Mars Glacier. The study site is in the northernmost valley, situated at an altitude of c. 270 m above the WGS-1984 datum. This valley has an enclosed surface hydrological catchment of 1.36 km$^2$ draining into the PICS.

**2. Methods**

The PICS site cannot easily be accessed overland so all analyses were based on remote sensing (cf. Warner et al., 2021). Initial characterisation was carried out using oblique aerial photographs taken from a British Antarctic Survey (BAS), DeHaviland Twin Otter aircraft in January 2011 and December 2018 (Fig. 2). Subsequent detailed assesment of the site was made during two overflights by the BAS aerogeophysical equipped Twin Otter VP-FBL on the 22$^{nd}$ and 30$^{th}$ December 2019 during transits to and from survey tasking for the International Thwaites Glacier Collaboration (ITGC).

The survey aircraft was equiped with a Riegl Q240i scanning LiDAR. The LiDAR data were processed by Terratec, who intergrated the raw scanning LiDAR data, Global Navigation Satellite System (GNSS) positional data and Inertial Navigation System (INS) attitude data, and carried out boresight calibration based on repeat passes over Rothera Research Station on Adelaide Island (67°34′06″S 68°07′33″W). The resulting point cloud provided acurate measurements of surface elevation on the 22nd and 30th December 2019 (~10 cm standard deviation) across the study area (Figs. 3a and b). The aircraft survey altitude meant that a point density of 0.2 to 0.4 points per $m^2$ was achieved. These elevation data were interpolated onto a 1 m mesh raster to allow visualisation and comparison between survey dates. The resulting very high-resolution data reveals how the surface elevation changed in the short (8 day) period between overflights.

In addition to raw elevation values, the LiDAR also returns reflection intensity; the strength of the LiDAR reflection at every point (Kashani et al., 2015). We carried out two simple adjustments to the raw intensity values: First, we corrected for the reduction in intensity due to the scan angle by fitting a 2nd order polynomial to a plot of scan angle vs intensity over a local area of aproximately flat white snow. Second, the amplitude was normalised to simplify comparison between the two flights. These corrections were not rigorously calibrated between flights, and additional corrections for range to ground and aircraft attitude have not been carried out. However, the general pattern of intensity with these basic adjustments provides useful data (Figs. 3c and d), with low intensity returns reflecting rough or wet regions, while higher intensity returns indicate a simple and more reflective surface (Kashani et al., 2015).

To assess the geomorphology of the subglacial bed we utilised data from a CReSIS 600-900 MHz accumulation radar (Arnold, 2020). This radar system was mounted on the survey aircraft during the December 2019 survey flights, and collected data simultaneously with the LiDAR. The data from the accumulation radar has an along track resolution of 20 to 30 m, depending on aircraft speed, and a depth resolution of c. 50 cm (Arnold, 2020). The data are plotted as the amplitude of the radar reflections on a log scale (Fig. 4). The vertical elevation was calculated assuming radar velocities of  300 m/us and 168 m/us in air and ice respectivley.

Longer term changes in PICS surface elevation were determined from the Reference Elevation Model of Antarctica (REMA; https://www.pgc.umn.edu/data/rema/) using data from 2012 to 2017 (Howat et al., 2019). REMA is a high resolution, time-stamped Digital Elevation Model (DEM) of Antarctica at 8-meter spatial resolution. The provided REMA dataset includes both an average mosaic DEM assembled from multiple strip DEMs and the underlying time-stamped 'strip' files. The strip files were generated by applying fully automated, stereo auto-correlation techniques to overlapping pairs of high-resolution optical satellite images, using the open source Surface Extraction from TIN-based Searchspace Minimization (SETSM) software (Howat et al., 2019). The strip files are not registered to satellite altimetry, meaning that although relative elevation within a strip is robust they have lower absolute accuracy. To counter this issue, and allow assesment of long term local changes in elevation, we used LiDAR data from an area of exposed rock as a fixed elevation reference (marked in Fig. 3a). Each satellite DEM was shifted to have approximately zero mean ofset in this reference area. Elevation strips were available in the study area for 2013/09/12, 2014/02/10, 2016/02/14 and 2017/01/17.

### 3. Results

### 3.1 Aerial photographs and regional setting

Oblique aerial photographs taken on 14th January 2011 show the PICS during, or shortly after, its initial formation (Figs. 2a and b). It consists of a depression in the glacier surface bounded by shear ice cliffs formed by the ongoing collapse large blocks of ice around its margins (Fig. 2b). On the down glacier side, the ice cliffs undercut intact glacier ice. Further down glacier there is a curvilinear deformation of the ice surface which continues southwards towards the glacier terminus. Later photographs of the PICS taken on 3rd December 2018

show further calving of blocks of ice have resulted in ongoing retreat of the bounding ice cliffs, expanding the size of the depression (block collapse, Fig. 2d). Drifting snow has partially obscured the cliffs on the northern side, but several concentric bridged crevasses oriented towards the PICS show further cliff failures in progress. LiDAR data show that the PICS had reached a size of ~280 by 350 m by 2019 (Fig. 3). There is no evidence in the 2018 photographs (Figs. 2c and 2d) of the down-glacier deformation of the ice surface apparent in 2011.

Other features present in 2018 include supraglacial meltwater steams forming multiple incisions in the ice cliffs on the southern side, and rock debris on the snow surface to the southwest where the supraglacial streams are concentrated. (Fig. 2d). A number of horizontal dust or rock layers can be seen within the ice cliff face in both the 2011 and 2018 images.

**3.2 LiDAR geomorphology and reflectivity**

The 2019 LiDAR data show the base of PICS is ~245 m above the WGS-1984 datum, and the surrounding ice surface is >270 m (Fig. 3a), indicating the PICS is currently ~25 m deep. The area of the PICS, enclosed by the ice cliffs, is 0.067 km$^2$. The base of PICS includes an assortment of ~ 2-3 m topographic highs particularly towards the centre. The deepest parts of the PICS (243.5 to 244 m) are immediately adjacent to the ice cliffs, forming an internal 'moat'. The ice cliffs are steepest to the south and west, with the 20 m high ice block calving

from the southern ice cliffs (Fig. 2d) clearly resolved in the LiDAR data (Fig. 3a and b). The detailed structures within the PICS were generally consistent between the two flights on the 22nd and 30th December 2019 (Figs. 3a and b). However the approximately 100 m wide 10 m deep catchment depression that extends southward from the PICS (between 260 and 270 m attitude, indicated by the shading of the topography on the 22nd of December flight in Fig. 3a) is not well resolved on the 30th of December (Fig. 3b), despite elevation values being recovered

at similar angles from the centre-line further along the LiDAR swath. This is result of a decline in LiDAR reflection intensity in this area between the two flights (Figs. 3c and d) with the area of low reflection intensity expanding along the entire southern margin of the PICS (Fig. 3d). This zone of low/absent reflectivity corresponds to the area traversed by numerous supra-glacial streams observed in the December 2018 aerial photographs (Fig. 2d) and is likely a result of an increase in the amount of supraglacial melt water in this area

between the two flights. The patches of distinct high and low reflectivity with abrupt linear edges south of the PICS on the 30th December 2019 flight correspond to areas of exposed rock (Fig. 3d).

**3.3 Subsurface geomorphology**

Radar transects flown in orthogonal directions across the depression (Fig. 1b) show significant clutter likely associated with off axis reflections and multiple reflections between the aircraft and the ice surface (Fig. 4).

However, shallow reflectors 35-50 m below the ice surface indicate that the PICS overlies a broad topographic bowl in the subglacial bed (Fig. 4b). However, a significant bedrock dam is not imaged on the down-glacier side of the PICS (Fig. 4d). Bright reflectors at depths of 30-50 m directly beneath the PICS, are hard to interpret and

may be due to the ice sheet bed, englacial water, or off axis reflectors. The base of Mars Glacier is imaged beneath ~400 m of ice (Figs. 4a and c).

**3.4 Surface elevation changes**

The REMA surface elevation measurements from 2013, 2014, 2016 and, 2017 (Fig. 5), and LiDAR from 22nd and 30th December 2019 (Fig. 3) show that the PICS has undergone dynamic changes in depth. In 2013 it had a maximum observed depth of ~44 m and a volume of ~2,796,000 m$^3$ (Fig. 5). Since 2013 the base of the PICS has been rising in all surveyed years, with ~1,405,000 m$^3$ of material infilling the PICS between 2013 and 2019. The closely spaced LiDAR observations on 22nd and 30th December 2019 show the PICS floor rose by ~1.18 m; equivalent to ~71,000 m$^3$ of meltwater input in just 8 days. As the PICS floor rose, ice in the surrounding catchment fell by 10 to >80 cm (Fig. 6), with the losses concentrated on the steep slopes to the west and south, including areas traversed by active supraglacial meltwater steams in the December 2018 aerial photographs (Fig. 2d). Local surface melting, if spread across the catchment, may account for a large part of the observed infilling of the PICS. This would require ~7 cm of surface melt across the local catchment in 8 days.

Cross sections of surface elevation changes in the depression (marked by blue lines in Fig. 6 and shown in Fig. 7), show the infilling of PICS since 2013. The N-S line shows the development of the drifting snow infill on the northern side of the depression and the collapse of the ice cliff on the southern side forming an ice block between the 2017 REMA and the 2019 LiDAR surveys, which has become detached and sunk into the depression. The W-E line also shows the snow infill on the northern side of the depression and the continued retreat and steepening of the ice cliffs to the west. It also crosses one of the raised features which are 2-3 m above the PICS floor. A plot of the mean elevation change extracted from the two profiles crossing the PICS shows a more or less linear PICS infill rate of 3.18 ma$^{-1}$ between 2013 and 2019 (Fig. 8). A substantially greater increase in elevation of 1.18 m recorded between the 22nd and 30th December 2019 LiDAR surveys equates to a short-lived PICS infill rate of 53 ma$^{-1}$.

**4. Discussion**

Collectively, the evidence presented here is consistent with PICS being the result of rapid drainage of a subglacial lake, and collapse of the overlying ice during, or before, January 2011. Collapse was via a subglacial drainage conduit under Mars Glacier which caused partial collapse or deformation of the glacier surface showing the path of the outflow channel (Fig. 2b). Although we can't rule it out, this deformation of the glacier surface is unlikely to have been formed by an outburst of supraglacial meltwater from the PICS as undeformed glacier ice remains between the assumed site of the subglacial drainage conduit and onset of the down glacier surface expression of the outflow (Fig. 2b). The radar transects identify a broad bedrock depression in orthogonal transects which could retain subglacial water (Fig. 4).

The rapid nature of the drainage event is supported by the presence of the steep bounding ice cliffs which suggest loss of hydraulic support for the overlying ice and its collapse into a subsurface void following drainage of the subglacial lake. This ice cliff formation has been an ongoing process shown by the detachment of a block of ice, and formation of a new ice cliff on the southern side of the depression between the 2017 REMA and the

2019 LiDAR surveys (Fig. 7), and the formation of the concentric crevasses which mark similar ongoing, or partial, structural failures of the ice on the northern side (Fig. 2d). The lake occupying the subglacial cavity must have been at least 46 m deep, using the maximum height of the ice cliffs in 2013 as a proxy for water depth. This adds to the inventory of deep subglacial water bodies under Antarctic Peninsula glaciers including Hodgson Lake with a water depth of 93.4 m (Hodgson et al., 2009b) and the subglacial lake under Crane Glacier with a water depth in 'the order of 100 m' (Scambos et al., 2011). However, unlike Hodgson Lake and the subglacial lake under Crane Glacier, the depth of the ice column between the surface depression and the underlying rock cavity is not known.

As the subglacial cavity is not constrained by a bedrock dam (Fig. 4d), its rapid drainage is consistent with the failure of a grounded ice dam at its lowest point, with water able to break through the seal and escape beneath Mars Glacier. This could occur as a result of a cold-based ice dam being eroded by surface meltwater reaching the bed (this process has also been described in volcanic and hydrothermal systems in Iceland, Björnsson, 2003), water level or pressure increases in the cavity exerting hydrostatic pressure on the ice, or floatation of an ice dam causing it to break contact with the bed triggering drainage. Glen's (1954) observations of glacier lake drainage events in British Columbia showed that this process can be asymmetric, with enlargement of the lake under the ice in the downhill direction causing increased pressure there relative to the other points, the opening of conduits to the subglacial hydrological network, and rapid emptying. Subglacial lake outburst flood models refer to this as 'super flotation water pressures' where excess water pressure exceeds the ice overburden pressure and drives water along the ice–bed interface, creating conduits linking into pre-existing subglacial drainage paths (Clarke, 2003).

As with conceptual models of (subaerial) glacial lake outburst floods, frictional heat dissipated by the escaping water can enlarge subglacial conduits allowing continued drainage (P. 92, Benn and Evans, 2010). In these models, water pressure eventually falls to the extent that closure of the conduit by ice creep exceeds melting rates and the seal is reformed allowing the lake to refill. If this mechanism is correct, we can assume that full, or at least partial resealing had occurred during, or before, 2013 when the lake began to refill, decreasing the depth of PICS by ~ 30 m in six years (Figs. 7 and 8). At some time between 2011(Figs. 2 a and b) and 2018 (Figs. 2 c and d) the surface expression of the subglacial outflow conduit disappeared which may indicate that it was no longer active.

The 1.18 m decrease in the depth of the PICS between 22nd and 30th December 2019 substantially exceeds the overall infill trend (Fig. 8) and coincides with a 32-year record-high surface melt in 2019/2020 recorded on the northern George VI Ice Shelf (Banwell et al., 2021). This suggests a dominant contribution from seasonal surface meltwater, whilst the relative contributions of englacial hydrological processes and snow accumulation remain unknown. The accumulation of blowing snow is considered minor as it is focused on the northern side of PICS (Fig. 2d). This pattern of rapid drainage and slow recharge has been described in subglacial lakes in Greenland (Willis et al., 2015; Livingstone et al., 2019; Liang et al., 2022).

Overall, the PICS feature conforms to Livingstone et al's. descriptions of 'water-filled cavities that drain rapidly beneath valley glaciers' (Fig. 7c in Livingstone et al., 2022; Willis et al., 2015; Liang et al., 2022). However, topographically analogous circular ice depressions, referred to as 'ice cauldrons' (or 'ice caldera') have also

been recognised in areas of volcanic or geothermal activity, for example in Iceland (e.g. Reynolds et al., 2019). Volcanic 'ice cauldrons' are caused by ice melting at their base and are likely to be linked to minor subglacial eruptions. They often occur in clusters that may trace the caldera margin, dyke or rift features. We consider a volcanic origin unlikely for the PICS. There is no evidence of local volcanism around Phobos Ridge or Mars Glacier, although the Beethoven Peninsula volcanic field (Fig. 1a) may extend towards the south eastern margins of Alexander Island (Smellie and Hole, 2021). Late Neogene post-subduction volcanism has been identified elsewhere on Alexander Island and southwest Palmer Land, likely related to the ascent and decompressional melting of mantle through slab windows. This encompasses widely scattered and isolated outcrops of the Late Neogene Bellingshausen Sea Volcanic Group between 8 – 2.5 Ma (Smellie et al., 1988). Thermal wavelength imagery of 700,000 year old volcanic rocks at Gluck Peak (148 km west) indicate elevated heat flow, which may be attributable to ongoing geothermal activity in Alexander Island (Smellie and Hole, 2021), although no primary volcanic landforms have been identified.

**4.1 Evidence elsewhere in Antarctica**

The feature described here is not unique. A similar but smaller 50 m diameter, 20 m deep feature within a wider 300 m depression was described in 1958 at Nobby Nunatak (63°25′S 56°59′W) near Hope Bay at the north eastern end of the Antarctic Peninsula (Koerner, 1964) (Figs.1 and 9a). Another 380 m diameter, 35 m deep depression was described in 1957 near Mount Wild and the Sjögren and Boydell Glaciers on the eastern Antarctic Peninsula (64°12'S, 58°58'W) (Aitkenhead, 1963) (Figs. 1 and 9b). Both features were referred to as 'ice caldera', but no link to volcanic activity was proposed. They occur in areas warmed by local föhn winds, raising temperatures above the freezing point, leading to formation of surface meltwater. Subsequent observations of the feature near Mt. Wild showed a rapid lake drainage event occurred between June 1961 and 2 August 1961 following a period of 'unusually warm temperature' resulting in the surface of the feature falling by c. 20 m and causing slabs of ice to collapse and form new ice cliffs around the rim. The author proposed that meltwater had accumulated both at the surface and in a subglacial bedrock basin and that drainage occurred via a subglacial channel.

More recently, in March 2022, and c. 1000 km further South an analogous ice collapse structure was photographed at Callisto Cliffs (71° 01' S., 68° 20' W; 92 km North of PICS on Alexander Island). This has characteristics consistent with drainage of another subglacial water body, including collapse of the overlying ice, and deformation of the glacier surface showing the down glacier path of the subglacial drainage conduit or surface meltwater (Fig. 10). Elsewhere in Antarctica, a 183 m × 220 m depression bounded by ice cliffs has been described in the Larsemann Hills (East Antarctica); although in this case the collapse of the overlying ice was caused by drainage of supraglacial water and nearby epiglacial lakes into an englacial cavity in 2017 causing its overfill and outburst (Boronina et al., 2021).

**4.2 Wider implications**

Whilst this feature and its analogues may simply be interesting glaciological anomalies, similar surface collapse features resulting from subglacial lake drainage events have been described from Greenland (Palmer et al., 2015; Willis et al., 2015; Howat et al., 2015). These examples all exhibit a hydraulic connection delivering

supraglacial meltwater to the bed of the ice sheet. Willis et al., (2015) suggest that this meltwater would be both warmer and have a lower viscosity which, combined with sporadic lake drainage events, could modulate and possibly accelerate downstream ice flow, with implications for ice sheet mass balance. Observational evidence of this has been provided by Lang et al (2022) who measured a transient threefold increase in glacier flow rates downstream of a lake discharge event in 2019. For subglacial lakes under the East Antarctic Ice Sheet, which are isolated from surface meltwater inputs, rapid drainage (and refilling) has been linked to a 10% acceleration (and subsequent deceleration) in ice flow (Stearns et al., 2008).

Our hypothesis is that these relatively rare subglacial lake drainage events, and the resulting ice collapse features, mark a threshold in the development of polythermal glacier behaviour. On the Antarctic Peninsula they provide a first surface expression of a supraglacial meltwater reaching the bed bringing about changes in basal hydrology in areas of regional warming. The two 'ice caldera' reported in the north eastern Antarctic Peninsula (Fig. 9) were first described in the late 1950's (Aitkenhead, 1963; Koerner, 1964). This followed the regional intensification of the Southern Hemisphere Westerly Winds from the 1920's (Perren et al., 2020) which likely increased the frequency of the föhn winds that cause rapid surface melt events in northeast Graham Land (Laffin et al., 2021), and coincides with the onset of regional warming of the north-eastern Antarctic Peninsula (data from Esperanza Station shown in Turner et al., 2016).

The spatial and temporal distribution of these features has not been systematically mapped and may therefore be underreported. However, the observation of these features in the northern Antarctic Peninsula in the late 1950's and the apparent recent appearance of similar features c. 1000 km further south on Alexander Island at the PICS and Callisto Cliffs may indicate a southward extension of this glacier response to regional warming. If delivery of supraglacial and englacial meltwater is responsible for the failure of the ice dam, followed by lake drainage through subglacial conduits and subsequent refill, it would indicate the activation or enhancement of hydrological networks. This would be consistent with work by Bowling et al. (2019) in Greenland which shows a shift from stable subglacial lakes above the Equilibrium Line Altitude (ELA) to hydrologically-active lakes near the ELA recharged by surface meltwater. Regional increases in Antarctic Peninsula surface melting have been reported by Abram et al., (2013), and Tuckett et al. (2019) show that delivery of surface meltwater to the bed of Antarctic Peninsula outlet glaciers triggers increases basal water pressure resulting in rapid seasonal ice flow accelerations (up to 100% greater than the annual mean). For glaciers discharging into ice shelves, ocean forcing has also been shown to influence seasonal flow rates at, and immediately inland of, the grounding line (Boxall et al., 2022).

The rapid drainage of the PICS subglacial lake and its subsequent refilling (largely) by surface meltwater provides a mechanism for transferring the impacts of a warming atmosphere and increased surface melting to the base of the ice sheet (cf. Willis et al., 2015). Under future + 3°C climate change scenarios (Deconto et al., 2021), enhanced volumes of supraglacial and englacial meltwater will be delivered to the bed of the region's glaciers further coupling surface climate processes to the bed. This will enhance subglacial hydrological activity and seasonal accelerations in ice flow can be anticipated for longer periods (cf. Hewitt, 2013), potentially influencing the deglaciation of the Antarctic Peninsula Ice Sheet.

**5. Conclusions**

Our detailed analysis of the changing geometry of a ~ 280 to 350 m wide circular, steep-sided depression in the surface of Mars Glacier is consient with the drainage of a c. 46 m deep subglacial lake and the collapse of the overlying ice. The collapse structure formed during, or prior to, 2011, and has been re-filling ever since,

reducing its depth from 45 to 25 m between 2013 and 2019. Although the long-term filling trend is approximately linear, repeat LiDAR observations separated by just 8 days in 2019 show the base of PICS floor rose by ~1.18 m, equating to ~71,000 m$^3$ of meltwater input from a catchment just 1.36 km$^2$. This coincides with the 32-year record-high surface melt in 2019/2020 recorded on the northern George VI Ice Shelf. The data therefore suggest that the subglacial hydrological networks in this region are immediately responding to changes

in surface climate. As subglacial hydrological networks have been linked to changes in glacier flow rates and grounding line retreat, this close coupling between the surface climate and the bed of the region's glaciers shows their ongoing vulnerability to regional warming.

**Data availability.** Data published in the NERC Polar Data Centre: Jordan, T., & Robinson, C. (2022). *Rectified*
*airborne Lidar data over Thwaites Glacier catchment between 1st January and 30th December 2019* (Version 1.0) [Data set]. NERC EDS UK Polar Data Centre. https://doi.org/10.5285/6909792B-FADF-4DE6-AC2A-D32FA76A8339

**Competing interests.** The authors have no conflicts of interest.

**Author contributions.** DAH and TAJ contibuted equally to this paper. DAH conceived the study from aerial photography, and NR provided and interpreted earlier aerial photographs of the collapse event. TAJ carried out the remote sensing acquisition and analysis. PTF sourced satellite images and TRR provided the geological

expertise. All authors contributed to the writing.

**Acknowledgements.** We thank the British Antarctic Survey (BAS) Operations Team and Air Unit for facilitating the remote sensing of the PICS site. Stephen Livingstone, one anonymous reviewer and the handling editor were generous with their knowledge, providing constructive advice and additional references which
strengthened our manuscript.

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

**Figures.**

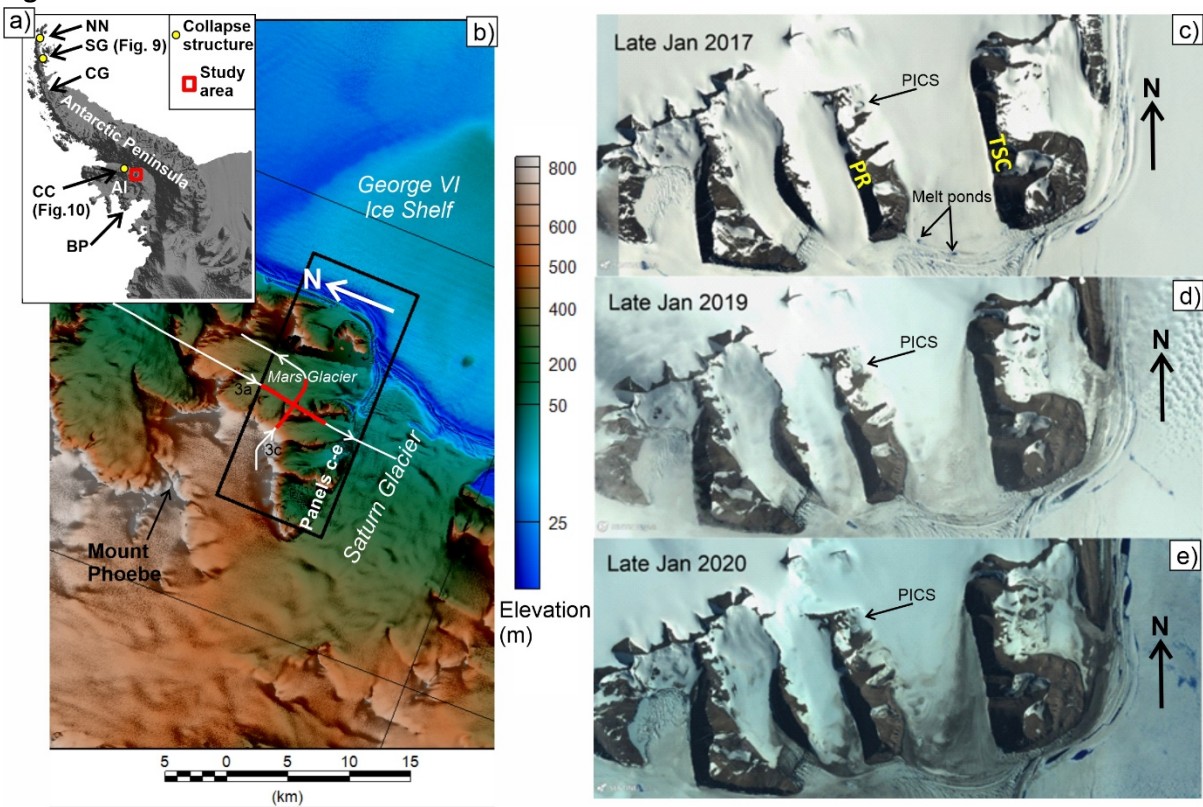

Figure 1: Location of Phobos Ice Collapse Structure (PICS). a) Location of Alexander Island (AI) and the PICS (red box) on the Antarctic Peninsula. Yellow dots mark other ice collapse structures on the northern Antarctic Peninsula at Nobby Nunatak (NN) (Koerner, 1964), and near Sjögren Glacier (SG) (Aitkenhead, 1963), and on the southern Antarctic Peninsula at Callisto Cliffs (CC) on Alexander Island. Other named features include Crane Glacier (CG) where satellite altimetry indicated draining of a subglacial lake between September 2004 and September 2005 (Scambos et al., 2011), and the Beethoven Peninsula volcanic field (BP). b) REMA digital elevation model of the study area (Howat et al., 2019). Thin white lines with arrows mark the 2019 LiDAR flights over the PICS, and red flight segments locate the radar sections in Fig. 4. Black box locates satellite images in panels c-e. c-e) Sentinel satellite images showing the PICS in 2017, 2019 and 2020.  Mars Glacier is bounded by Phobos Ridge (PR) to the west and Two Step Cliffs (TSC) to the east. Note evidence for extensive surface meltwater on George VI Ice Shelf, at the shear zone of Saturn Glacier and George VI Ice Shelf, and at the terminus of Mars Glacier in all years, and the increase in dust on the surface of the glacier in 2019 and 2020. Copernicus Sentinel data 2021, processed by ESA.

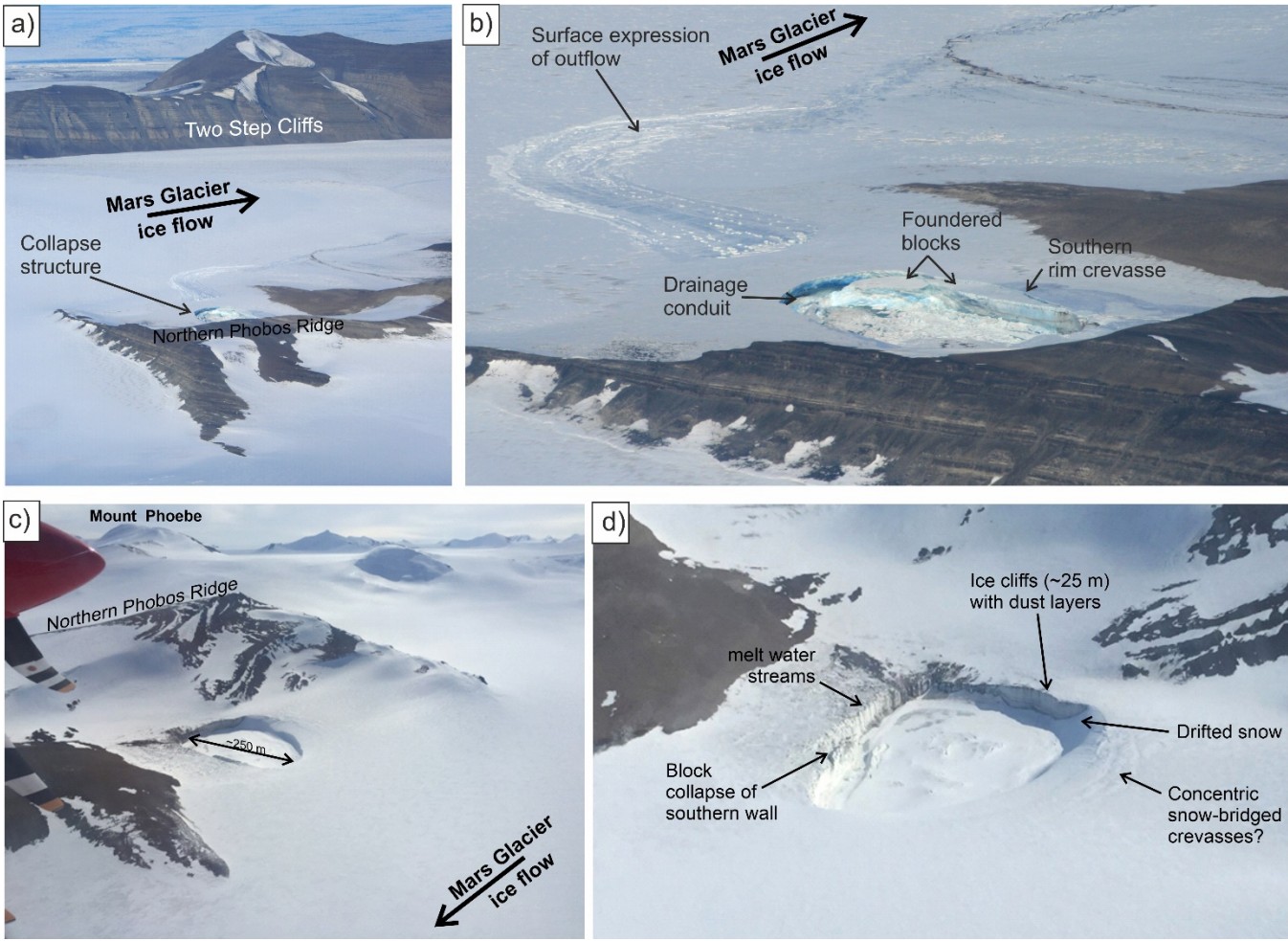

**Figure 2: Oblique aerial photographs of the PICS. a)** Overview of northern Phobos Ridge, Mars Glacier and Two Step Cliffs showing the location of the PICS on 14th January 2011. **a)** A closer view of PICS on the same date showing blocks of ice collapsing into the cavity forming ice cliffs, undercutting of the ice cliff in the (presumed) location of the subglacial drainage conduit, and the surface expression of the subglacial outflow channel heading down-glacier. Note the intact glacier ice between the assumed location of the PICS drainage conduit and the surface expression of the subglacial outflow channel on Mars Glacier. **c)** PICS on 3rd December 2018 looking northwest from Mars Glacier towards Phobos Ridge and Mount Phoebe beyond, and **d)** a closer view on the same date looking approximately southwest. Note the expansion of the feature (further block collapse on the southern side, formation of concentric crevasses on the northern side; left and right of image respectively), the presence of meltwater streams flowing into PICS via the ice cliffs, and the absence of the surface expression of the subglacial outflow channel on Mars Glacier. Author photographs (NR and DAH).

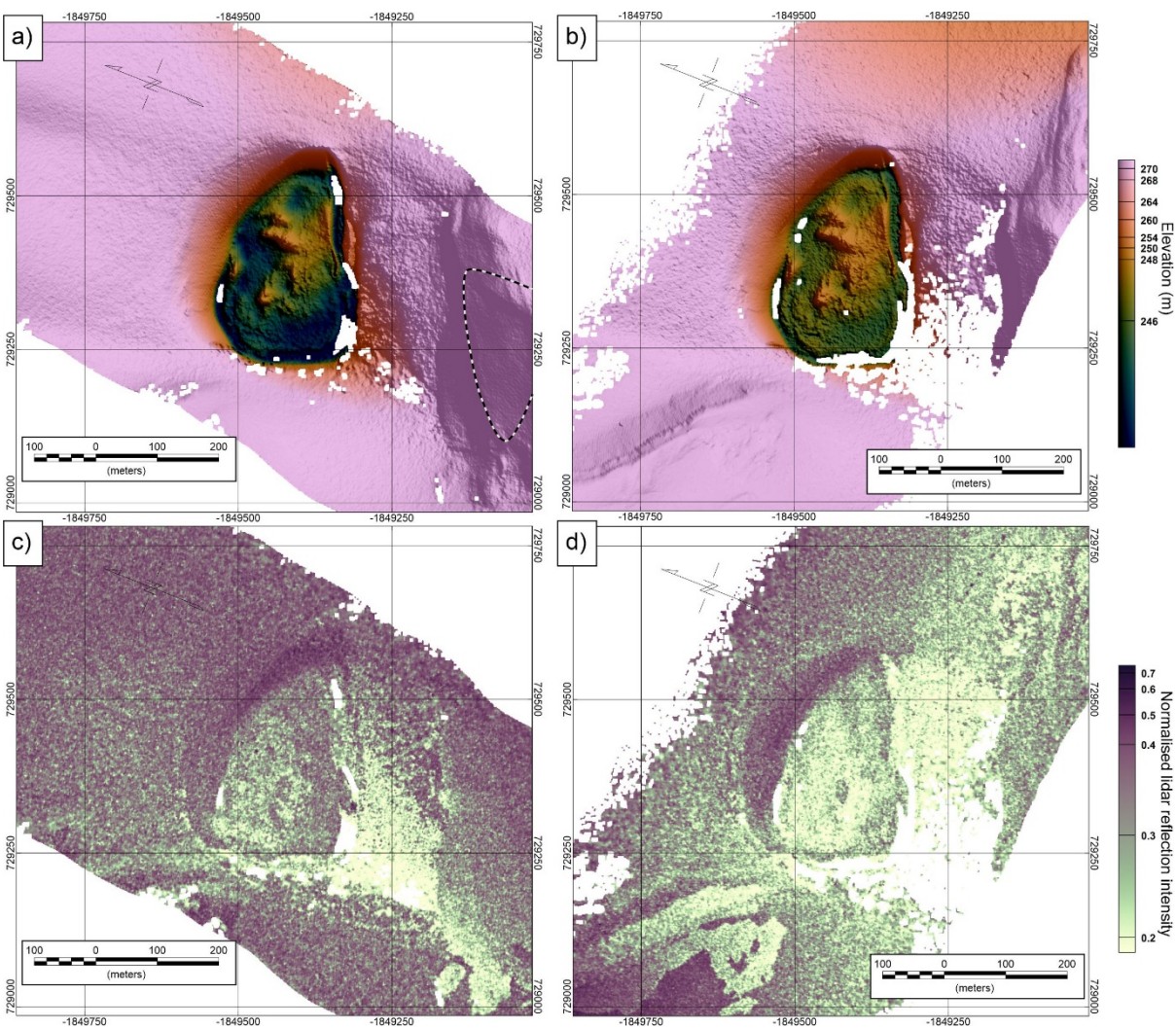

**Figure 3: LiDAR imaging of PICS. a) LiDAR elevation from flight on 22nd December 2019. Black and white dashed line outlines the area of bedrock used as a reference elevation to correct elevation of satellite strip images in Figs. 5 and 6. b) LiDAR surface elevation from flight on 30th December 2019. c) LiDAR reflection intensity from 22nd December 2019 flight. d) LiDAR reflection intensity from 30th December 2019 flight. Note the catchment depression extending southward from the PICS imaged on 22nd of December (bottom right of images) had low/absent reflection intensity on 30th of December due to the presence of numerous supraglacial meltwater streams.**

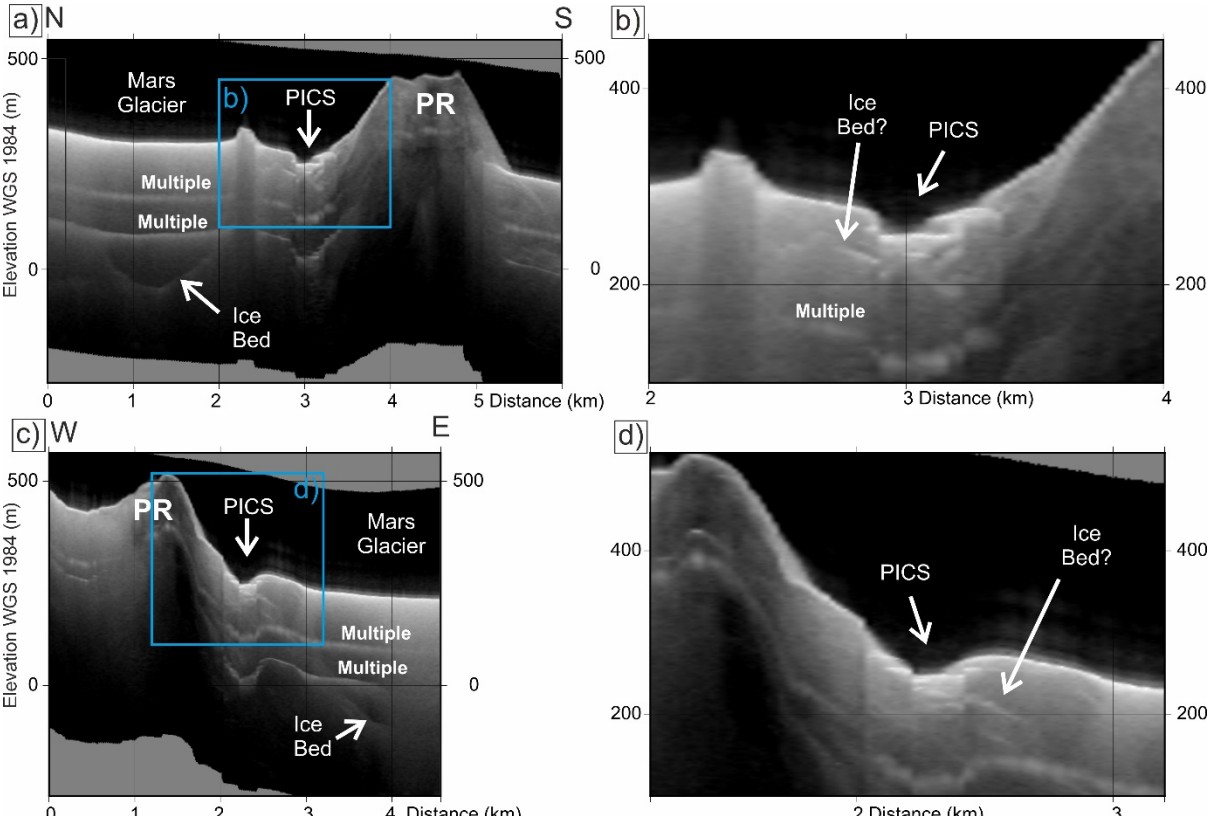

**Figure 4: Radar sections along the red flight lines shown in Fig. 1b. a) North to South section crossing Mars Glacier and Phobos Ridge (PR). Note ice bed reflection ~350 m below the surface of Mars Glacier. b) Detail of the North to South section over PICS (located by the blue box in panel a). Note potential ice-bed reflection between 35 and 50 m beneath the ice surface. c) West to East section. d) Detail of PICS on the West to East section (located by the blue box in panel c). Image brightness indicates strength of the returned reflection on a log scale.**

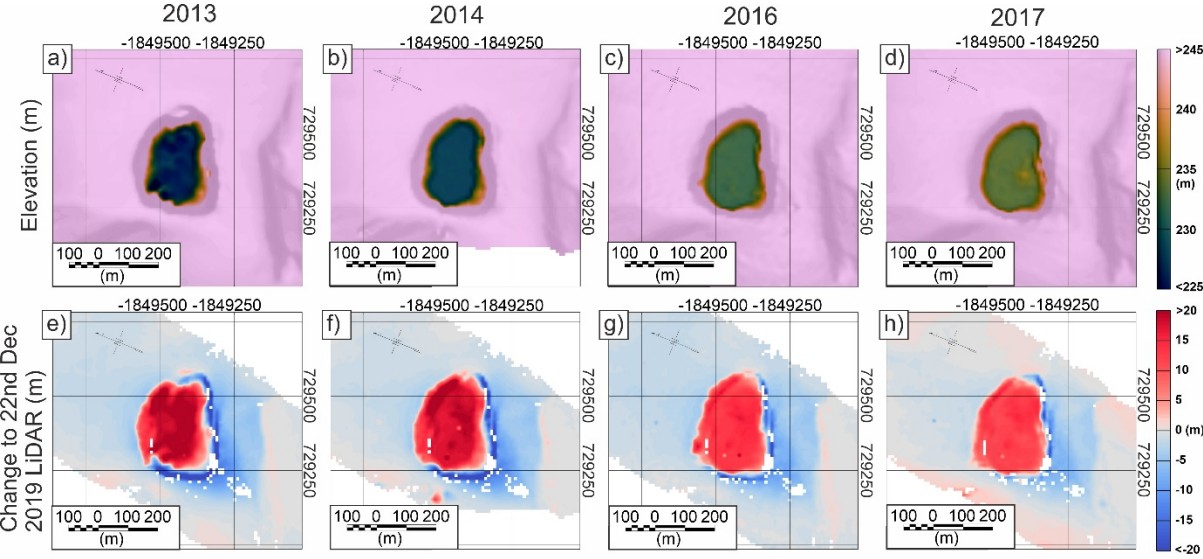

**Figure 5: Long term surface elevation changes from REMA strip DEM's over the PICS. Panels a-d show REMA DEM's from 2013 to 2017, corrected to the area of reference elevation shown in Fig. 3a. Note uniform colour scale, clipped to accentuate variations in the elevation of the PICS floor. Panels e-h show elevation difference between strip DEM's and December 22nd LiDAR elevation grid – positive values show areas where elevation has increased with time as the cavity has refiled, and negative values show areas of the catchment where elevation has decreased as a (likely) result of surface melt. Note deep and more rugged floor of the PICS in 2013 which suggests either that drainage may have been complete, allowing the basal topography to show through, or more likely that the raised features are fragments of collapsed ice cliff, or ice blisters (Moore, 1993). The very flat base of the PICS in subsequent years would be consistent with progressive filling of the cavity, and floatation of ice off the bed or the melting or incorporation of these features.**

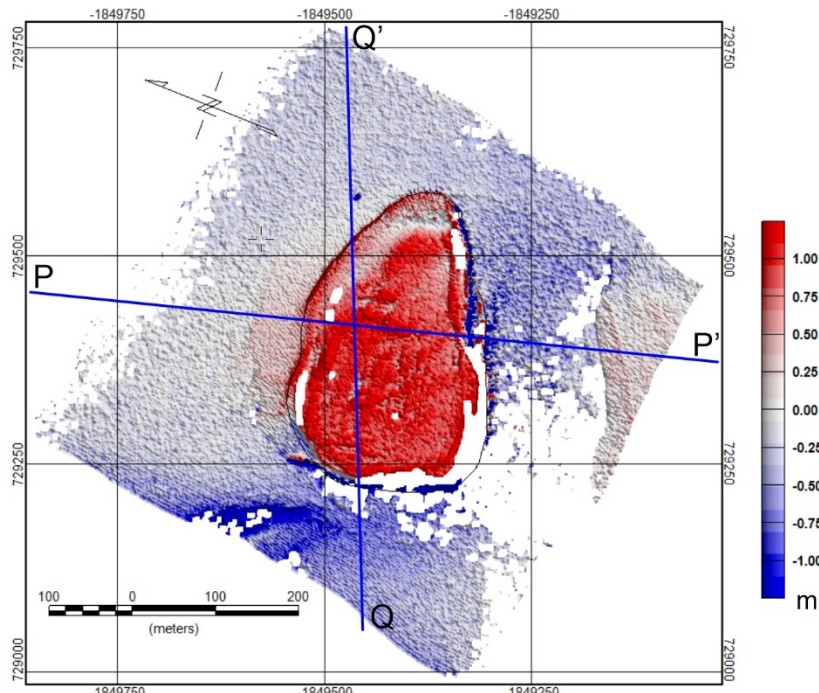

**Figure 6: Change in surface elevation between 22<sup>nd</sup> and 30<sup>th</sup> December 2019. Blue lines locate section profiles shown in Fig. 7.**

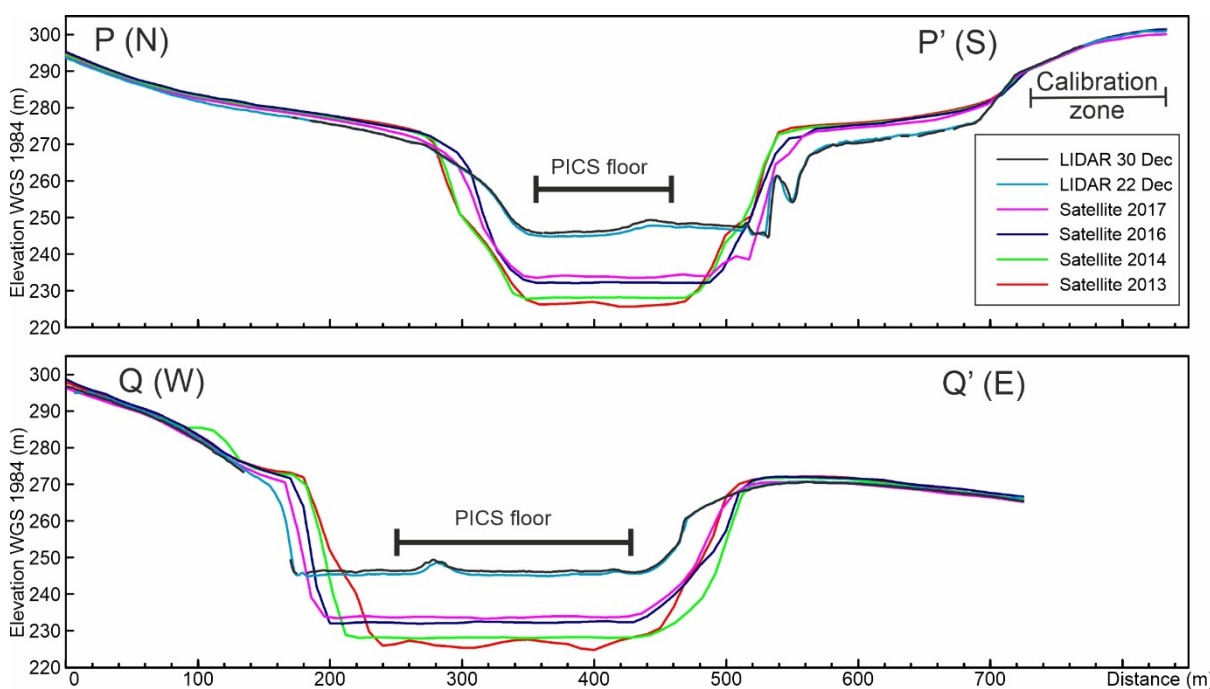

**Figure 7: Profiles of surface elevation changes over the PICS structure (locations of P-P' and Q-Q' shown in Fig. 6). The Reference elevation area (Fig. 3a) was used to calibrate the mean vertical shift applied to each satellite derived REMA strip DEM (2013 +1.93 m, 2014 +0.87 m, 2016 +3.12 m, 2017 +1.39 m). 2019 LiDAR profiles are un-adjusted. The 'PICS floor' region was used to calculate the mean elevation of the base of PICS plotted in Fig. 8. Note the block collapse in the 2019 LiDAR data (upper panel) and the formation of a new ice cliff.**

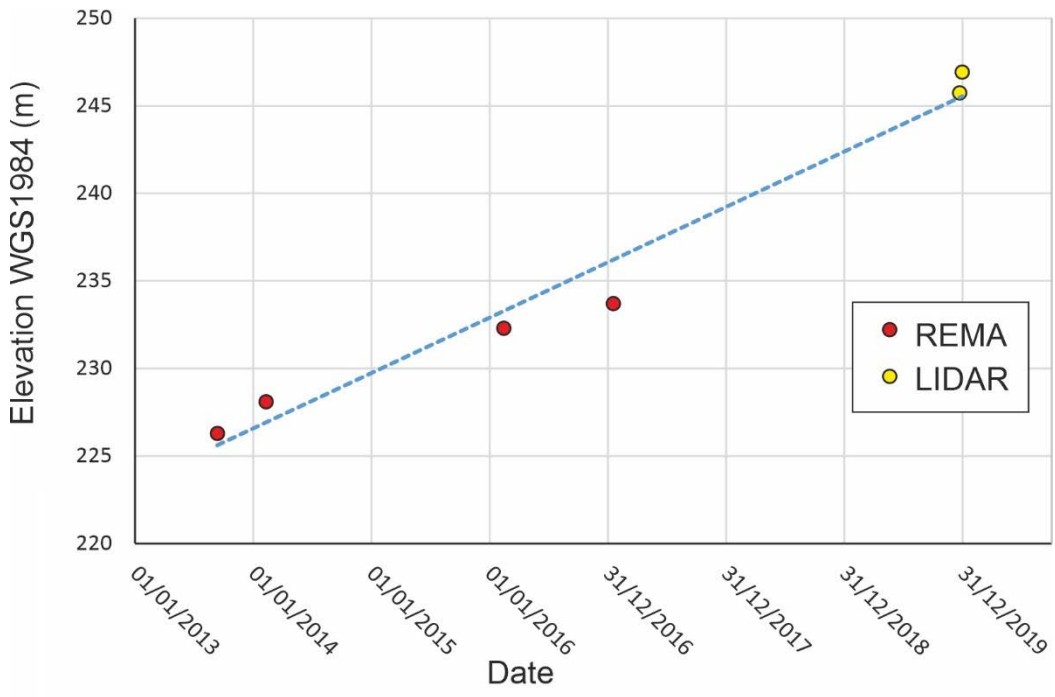

**Figure 8: Mean elevation of PICS floor with time, based on profiles of satellite and LiDAR DEM's in Fig. 7. Slope of best-fit line indicates a mean increase of 3.18 m per year.**

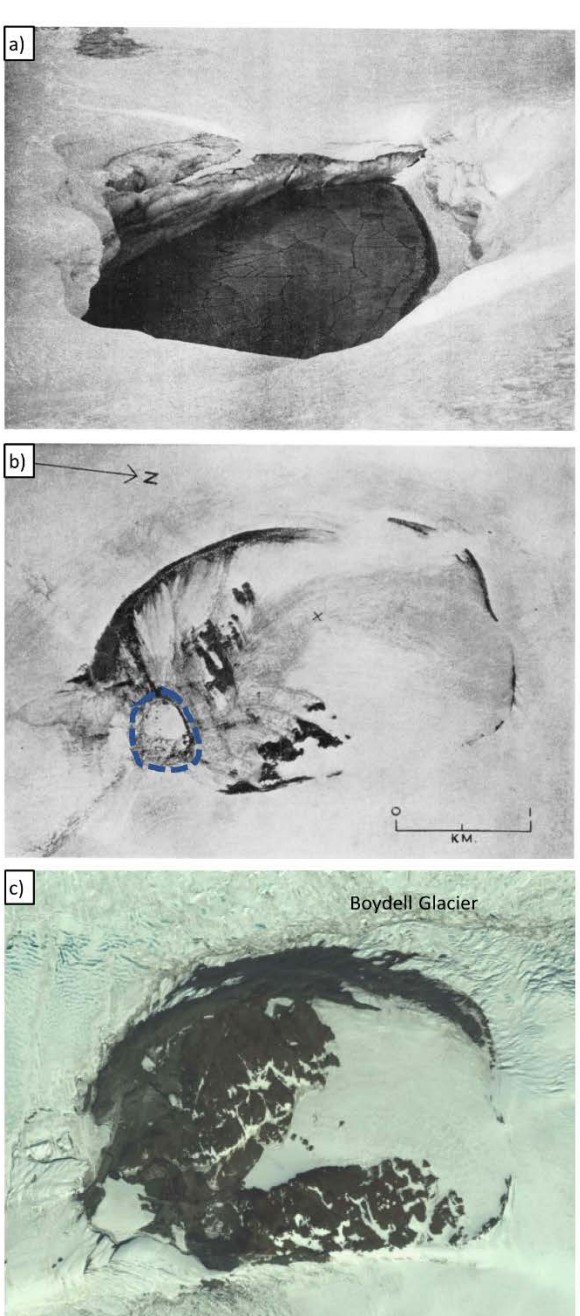

**Figure 9: Analogous ice collapse structures on the northeastern Antarctic Peninsula. a) The 'Ice caldera' at Nobby Nunatak, Hope Bay in July 1958 (Koerner, 1964). This was approximately 30-50 m in diameter with 20 m high ice cliffs and partially filled with water when the photograph was taken. The feature had disappeared by 1960. b) The 'Ice caldera' (marked by blue dotted line) 2.8 km west of Mount Wild near Sjögren and Boydell Glaciers in February 1957 (Aitkenhead, 1963). This feature was 380 m diameter feature with 20 m ice cliffs. c) The same feature in 2015 (shortly before the retreat of the Boydell Glacier). With the loss of Prince Gustav Channel Ice Shelf between 1989 and 1995 (Cooper, 1997) the bounding glaciers have retreated, and the feature is now part of an extensively deglaciated nunatak adjacent to the coast of Sjögren Inlet. Google Earth Imagery date 1/1/15, Maxar Technologies.**

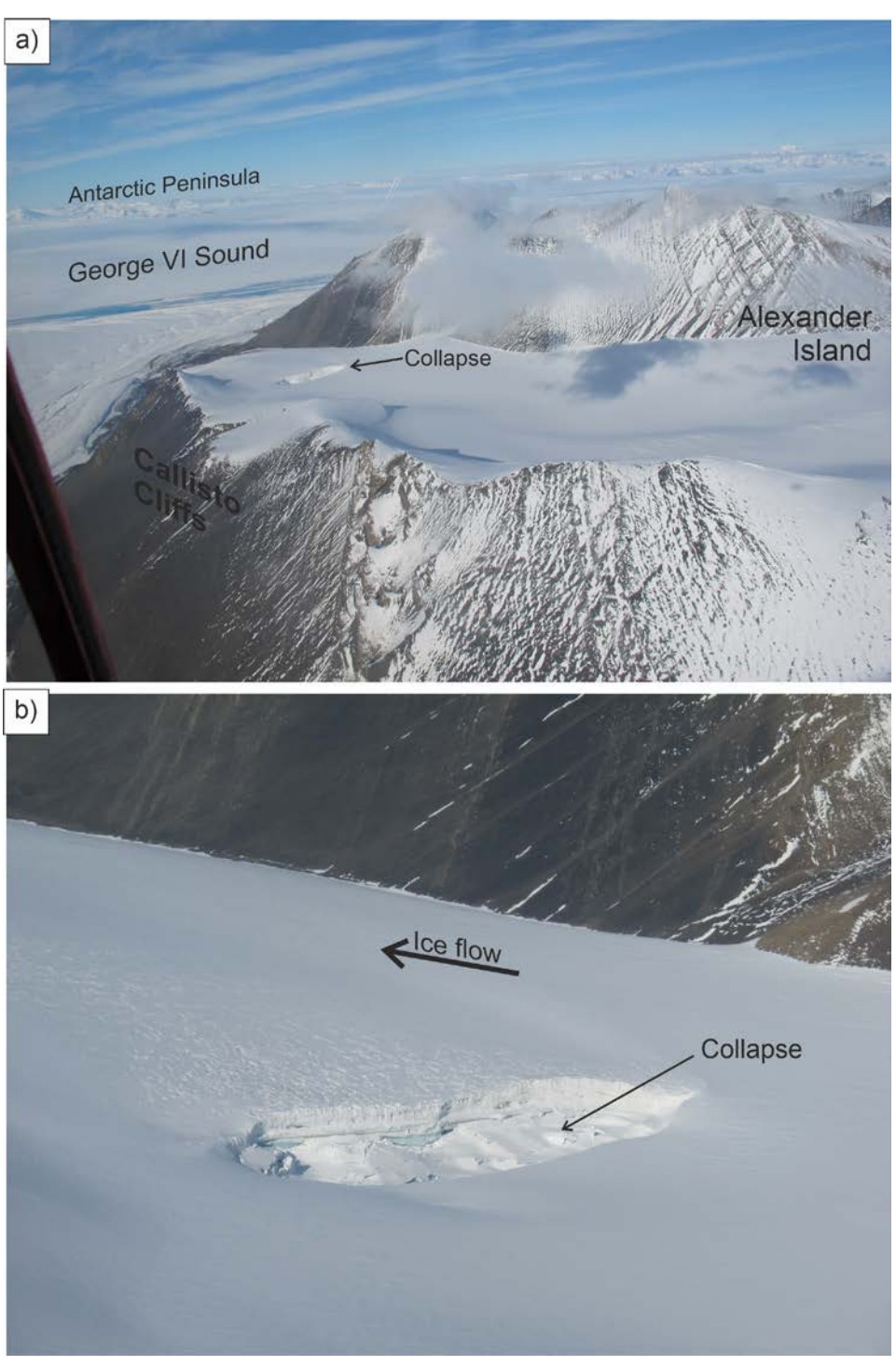

**Figure 10. Ice collapse structure at Callisto Cliffs on Alexander Island first observed in March 2022. a) View looking approximately southeast towards George VI Sound. b) Zoom on collapse structure. Note extensive surface crevassing on the down-stream side of the structure which may be the result of supraglacial meltwater or deformation of the surface over an outflow conduit.**

END