# Peer review of "Drainage and refill of an Antarctic Peninsula subglacial lake reveals an active subglacial hydrological network"

_The Cryosphere, 2022_

## Referee Comment (RC1)

**Drainage and refill of an Antarctic Peninsula subglacial lake reveals and active subglacial hydrological network. Hodgson et al. 2022. TCD**

General Comments

I enjoyed reading this paper. It is well written with excellent supporting figures and provides an important contribution to our understanding of smaller active subglacial lakes in the Antarctic Peninsula, and further support that surface meltwater is reaching the bed above the grounding line. The paper uses remote sensing and aerogeophysical platforms to characterise the ice collapse structure and the slow refill of the lake. Further examples of ice collapse structures are identified around the Peninsula suggesting these lakes are rare but not isolated phenomena. I am therefore in favour of this being published. However, I do have some general comments/ questions summarised below and more specific line-by-line edits and comments:

1. **Constraining/ inferring rapid subglacial lake drainage.** The rapidity of drainage seems to be largely inferred from the steep ice walls. Could this also be a result of other glaciological factors (e.g., ice thickness, crevassing …)? Some discussion or further support for rapid drainage would really help the paper. In particular, could you use the satellite archive to look for evidence of when the lake drained, and over what time-span? This would really strengthen the argument. Out of interest, the surface collapse pattern reminds me of the drainage of englacial Lake Dålk (see Fig. 2 – of Boronina et al., 2021), which also drained rapidly but with a thin ice lid.

2. **Supraglacial meltwater inputs.** There might not be the resolution to do so, but I wonder if there is a simple back-of-the envelope calculation that could be done to evaluate the contribution of surface meltwater to the uplift rates based on cumulative modelled catchment runoff between 2013-2019? There is a nice study by Liang et al. (2022 – TC) who correlate lake recharge with surface melt inputs in Greenland.

3. **Lake drainage trigger?** Just a thought, but if the ice is relatively thin, could the lake have been trapped by cold based ice? Subsequent cryohydrological warming by surface water getting to the bed, might then be impacting these seals around the Peninsula?

4. **Wider implications.** I like the idea that these subglacial lake drainages might be the result of climate warming, even though it is a bit speculative. And certainly, I buy the idea that surface melt getting to the bed could start to trigger subglacial lake drainage and activate hydrological networks, which is consistent with, for example the Tuckett et al. (2019) paper, and some of the ideas in Bowling et al., (2019) for Greenland (i.e. that lake could activate as the ELA rises). But there is some circularity in your argument as written, as you use the Boxall et al. (2022) paper to support the idea of subglacial hydrological networks (L255) but then suggest the identification of active subglacial hydrological networks supports the Boxall et al. (2022) results. I think the argument would be stronger if you used your data as evidence for surface meltwater getting to the bed of the ice sheet and influencing subglacial hydrological processes; and then use that to support the idea that the seasonality in ice flow that has been observed can be explained by surface melt above the grounding line and not ocean forcing.

Boronina, A., Popov, S., Pryakhina, G., Chetverova, A., Ryzhova, E. and Grigoreva, S., 2021. Formation of a large ice depression on Dålk Glacier (Larsemann Hills, East Antarctica) caused by the rapid drainage of an englacial cavity. Journal of Glaciology, 67(266), pp.1121-1136.

Bowling, J.S., Livingstone, S.J., Sole, A.J. and Chu, W., 2019. Distribution and dynamics of Greenland subglacial lakes. Nature communications, 10(1), pp.1-11.

Liang, Q., Xiao, W., Howat, I., Cheng, X., Hui, F., Chen, Z., Jiang, M. and Zheng, L., 2022. Filling and drainage of a subglacial lake beneath the Flade Isblink ice cap, northeast Greenland. The Cryosphere Discussions, pp.1-17.

- Stephen Livingstone

Specific Comments

L6-7 – I am not sure this is strictly true because the sample size of ice velocity measurements in response to lake drainages is still so small. Certainly, a lot more is known about the hydrological activity of subglacial lakes, so could just remove the dynamic part of the sentence.

L9 – You certainly infer a rapid drainage, but I am not sure you describe it as you only know it occurred pre-2013. I would suggest rephrasing slightly (or see general comment).

L20 – I would suggest modifying to: "including short-term accelerations in ice flow" for this reference as it focuses on seasonal patterns.

L21 – it sounds a bit odd to refer to airborne radio-echo sounding here when talking about ice surface elevation changes.

L33-36 – could cite Kingslake et al. (2017) to support this statement:

Kingslake, J., Ely, J.C., Das, I. and Bell, R.E., 2017. Widespread movement of meltwater onto and across Antarctic ice shelves. Nature, 544(7650), pp.349-352.

L38 – ice shelf – two words.

L41 – Lai et al. look specifically at the vulnerability of ice shelves rather than specific collapses. Might be better to cite a study that has looked at collapse of an Antarctic Peninsula ice shelf and the influence of meltwater, e.g., van den Broeke (2005).

van den Broeke, M., 2005. Strong surface melting preceded collapse of Antarctic Peninsula ice shelf. Geophysical Research Letters, 32(12).

L73 – can you state the resolution you gridded the data at here?

L80 – Do you mean "could not be carried out", given the "However" at the start of the next sentence?

L101 – Is this fixed surface elevation based on bedrock outcrops? It would be useful to clarify how you did this.

L122 – Would help if this depression was mentioned in the caption or annotated in Figure 3a.

L141 – Can you rule out the reflector being the bed based on the depth of the PICS in 2013 (i.e., an extra ~30 m below the 2019 LiDAR data)?

L157 – Could some of the variation between the northern and southern cliff sections be associated with ice flow?

L164 – I found it confusing that you talk about a decrease in basin depth in one sentence and then elevation increase in the other. You could be consistent across terms here. A space is also needed between 1.18 and m.

L180 – "This suggests either that drainage…"

L191 – The flotation of the ice dam could also have caused an initial sheet flood that then developed into channels, as suggested for some Icelandic subglacial lake drainages based on the hydrographs and modelling – e.g., Flowers et al. (2004)

Flowers, G.E., Björnsson, H., Pálsson, F. and Clarke, G.K., 2004. A coupled sheet-conduit mechanism for jökulhlaup propagation. Geophysical research letters, 31(5).

L228 – It is not clear what "subsequent observations" you are referring to here? Could do with a citation or link to a figure.

L247-250 – This is a long sentence and the final part is poorly phrased. Maybe end as: "… while Prince Gustav Ice Shelf, which they formerly discharged into, has disappeared…"

L254 – Delete "of"

L255 – I find this sentence to be rather misleading – it is a ~15% austral summer speed-up relative to background, and not an increase through time (I am not sure if there is the temporal resolution to determine whether this seasonal velocity response has been happening over a longer time period). And although it could well be surface melt, the Boxall paper also suggest that these changes could be forced by the ocean.

L255-256 – This sentence seems a bit out of place to me. It could be misleading in suggesting you equate your lake drainages with the seasonal speed-up.

Figures & Tables

Figure 1 – I struggle to see most of the yellow dots in panel a. Maybe need a black ring or a colour change. The colour is also a bit confusing against the elevation scale bar (all looks very low elevation). Maybe a simple outline would help with both the above points?

Figure 2 – these are amazing images!

Figure 5 – I am not sure the elevation colour scale helps here as the depression just looks black, with variations induced by the hillshade (I think?). Perhaps remove the hillshade, and consider changing the colour scale to better identify changes. Or you could produce difference maps relative to the 2013 result for 2014, 2016 and 2017.

Figure 9 – It would be helpful to have a dotted line showing the rough outline of the depression; I struggled to place it in panel (a).

---

## Author Response (AR1)

**Authors' point by point response to comments**

We have substantially revised the manuscript and figures based on the constructive comments of the reviewers. The revisions can be seen as track changes in the accompanying Word file.

One of the key changes is the addition of a new co-author (Neil Ross, Newcastle University) who provided us with aerial photographs of the site on, or just before its formation January 2011. These photographs, and Neil Ross's observations at the time, helped us describe the mode of formation in much more detail than was possible from the earlier satellite images.

We respond to the reviewers' comments below:

**Reviewer 1**

1. Constraining/ inferring rapid subglacial lake drainage. The rapidity of drainage seems to be largely inferred from the steep ice walls. Could this also be a result of other glaciological factors (e.g., ice thickness, crevassing ...)? Some discussion or further support for rapid drainage would really help the paper. In particular, could you use the satellite archive to look for evidence of when the lake drained, and over what time-span? This would really strengthen the argument….

Response: We now have images of the site in January 2011 (Figs. 2a and b) which show the feature during, or just after the collapse event. It also shows surface deformation of the downstream ice revealing the path of the subglacial drainage channel. This latter feature was subsequently infilled by snow and is not visible in later imagery. The photograph shows the ice cliffs in the process of collapse into the subglacial cavity due to loss of structural support. Further blocks of ice have collapsed off the ice cliffs in subsequent years (Figs. 2d and 7).

2. There might not be the resolution to do so, but I wonder if there is a simple back-of-the envelope calculation that could be done to evaluate the contribution of surface meltwater to the uplift rates based on cumulative modelled catchment runoff between 2013-2019? There is a nice study by Liang et al. (2022 – TC) who correlate lake recharge with surface melt inputs in Greenland.

Response: We have no data to separate meltwater inputs from the catchment, glacier surface, seepage through porous firn or a linked cavity network. However, the 1.18 m infill of the PICS between $22^{nd}$ and $30^{th}$ December 2019, at the same time as the catchment fell by 10 to >80 cm (Fig. 6), indicates a likely dominance of surface meltwater inputs to the infill. This coincides with the 32-year record-high surface melt in 2019/2020 recorded on the northern George VI Ice Shelf reported by Banwell et al., 2021. The Liang paper, and the Willis (2015) paper that it cites both provide useful analogues from Greenland which we have incorporated into the discussion.

3. Lake drainage trigger? Just a thought, but if the ice is relatively thin, could the lake have been trapped by cold based ice? Subsequent cryohydrological warming by surface water getting to the bed, might then be impacting these seals around the Peninsula?

Response: Yes – this is possible. We have expanded on this in the revised text.

4. Wider implications. I like the idea that these subglacial lake drainages might be the result of climate warming, even though it is a bit speculative. And certainly, I buy the idea that surface melt getting to the bed could start to trigger subglacial lake drainage and activate hydrological networks, which is consistent with, for example the Tuckett et al. (2019) paper, and some of the ideas in Bowling et al., (2019) for Greenland (i.e. that lake could activate as the ELA rises). But there is some circularity in your argument as written…

Response: We have rewritten this section with a greater emphasis on surface meltwater getting to the bed of the ice sheet and influencing subglacial hydrological processes. We have also removed the circularity and separated the statements on the role of surface melt vs. ocean forcing.

5. Specific Comments

Response: We have addressed each of the specific comments and other minor observations. Notable ones include:

'L191 – The flotation of the ice dam could also have caused an initial sheet flood that then developed into channels, as suggested for some Icelandic subglacial lake drainages based on the hydrographs and modelling – e.g., Flowers et al. (2004)'.

Response: This mechanism is unlikely now that we have evidence of the pathway of the outflow conduit (Figs. 2 a and b).

'Figures & Tables'

Response: Requested changes made to Figure 1. Figure 5 and Figure 9 have been redrawn as suggested.

**Reviewer 2**

1. Are there any accessible datasets which could be used to investigate the actual lake drainage prior to 2013?

Response: Yes - we now have images of the site in January 2011 (Figs. 2a and b) which show the feature during, or just after the collapse event. We have revised the site description and interpretations based on these new images. See response to Reviewer 1.

2. The Authors state that a southerly shift of subglacial lake appearance can be related to a changing glacier regime on the Antarctic Peninsula. This might be true, however, how can the Authors be sure that no subglacial lakes exist in these areas prior to 2013?

Response: We are careful to state that the drainage '*may* indicate the southward expression of this phase of glacier response to regional warming'; i.e., this is speculation. We have not observed similar drainage features elsewhere at these latitudes, but this may be due to under reporting. To address

this, we add the sentence: 'However, the presence of these features has not been systematically mapped and may therefore be underreported.'

3. Specific comments

Response: We have addressed these constructive observations in the revised version. The suggested additional references were really helpful and were used to strengthen the discussion. Notable comments include:

'Are there any accessible datasets which could be used to investigate the actual lake drainage prior to 2013?'

Response: The best evidence we have found are the aerial photographs from 2011 presented in Figs. 2a and b.

'L100-103: are there any rock outcrops covered by the LiDAR and/or DEM data that can be used for vertical and horizontal alignment and/or validation of the datasets?'

Response: yes – this has been carried out – see revised Figure 7.

'Fig. 5: I think consecutive DEM differences would be more interesting than the single time steps shown here.'

Response: The Figure has been redrawn as suggested.

'Are the REMA strips covering the hydrological catchment?

Response: Yes – those parts of the catchments that have good data are now represented in Fig. 5.